# Compressive stress triggers fibroblasts spreading over cancer cells to generate carcinoma in situ organization

Fabien Bertillot[1,9,10], Laetitia Andrique[2,3,4,10], Carlos Ureña Martin [5,10], Olivier Zajac[1], Ludmilla de Plater[6], Michael M. Norton [4], Aurélien Richard[2,3,4], Kevin Alessandri[1], Basile G. Gurchenkov[1], Florian Fage[7], Atef Asnacios [7], Christophe Lamaze [5], Moumita Das[8], Jean-Léon Maître [6], Pierre Nassoy [2,3,11 ✉] & Danijela Matic Vignjevic [1,11 ✉]

At the early stage of tumor progression, fibroblasts are located at the outer edges of the tumor, forming an encasing layer around it. In this work, we have developed a 3D in vitro model where fibroblasts' layout resembles the structure seen in carcinoma in situ. We use a microfluidic encapsulation technology to co-culture fibroblasts and cancer cells within hollow, permeable, and elastic alginate shells. We find that in the absence of spatial constraint, fibroblasts and cancer cells do not mix but segregate into distinct aggregates composed of individual cell types. However, upon confinement, fibroblasts enwrap cancer cell spheroid. Using a combination of biophysical methods and live imaging, we find that buildup of compressive stress is required to induce fibroblasts spreading over the aggregates of tumor cells. We propose that compressive stress generated by the tumor growth might be a mechanism that prompts fibroblasts to form a capsule around the tumor.

Cancer progression is a multistep process that involves tumor growth and dissemination of cancer cells through the body. The tumor microenvironment, the extracellular matrix (ECM), and different cell types such as immune cells, blood vessels, and cancer-associated fibroblasts (CAFs)[1,2] play an essential role in cancer progression. CAFs stimulate the proliferation and invasion of cancer cells[3–8]. During tumor progression, CAFs accumulate in tumors[5] and produce an excess of ECM[9,10], thereby forming a capsule that encloses the cancer cells[11–14]. This capsule can function as a barrier that limits tumor growth and results in the increase of pressure within the tumor[14,15]. Thus, at the early stage of tumor progression, before the onset of invasion, cancer cells and CAFs are spatially segregated. However, the current 3D in vitro models based on the co-culture of cancer cells and fibroblasts fail to recapitulate this characteristic organization, mainly because cancer cells and fibroblasts do not adhere to one another and segregate into independent spheroids.

Thus, it is unclear how this peculiar organization of early-stage tumors with CAFs surrounding cancer cells is achieved.

Here, we investigated if cancer cells and fibroblasts can self-organize or if external cues are required. Using biophysical methods and live imaging, we found that fibroblasts do not autonomously envelop cancer cells. Instead, confinement and further buildup of compressive stress are necessary to induce fibroblasts to spread over tumor cell aggregates. Altogether, these data, supported by a simple theoretical energetic picture, led us to propose that confinement and compressive stress generated by tumor growth are a prerequisite for CAFs to enwrap the tumor as observed in vivo.

## Results

### Spatial confinement is required to induce and maintain fibroblasts organization around cancer cells

To generate a 3D model that recapitulates early tumor stages, we encapsulated suspended cultures of colon cancer cell line HT29 and NIH3T3

[1]Institut Curie, PSL Research University, CNRS UMR 144, F-75005 Paris, France. [2]LP2N, Laboratoire Photonique Numérique et Nanosciences, Univ. Bordeaux, F-33400 Talence, France. [3]Institut d'Optique Graduate School & CNRS UMR 5298, F-33400 Talence, France. [4]VoxCell, TBM-Core, CNRS UMS 3427 & INSERM US 005, Univ. Bordeaux, F-33000 Bordeaux, France. [5]Institut Curie, PSL Research University, CNRS UMR3666-INSERM U1143, F-75005 Paris, France. [6]Institut Curie, PSL Research University, U934/UMR3215, F-75005 Paris, France. [7]Laboratoire Matière et Systèmes Complexes, Université Paris Cité, CNRS UMR7057, F-75013 Paris, France. [8]Rochester Institute of Technology, Rochester, NY, USA. [9]Present address: Department of Cell and Tissue Dynamics, Max Planck Institute for Molecular Biomedicine, 48149 Münster, Germany. [10]These authors contributed equally: Fabien Bertillot, Laetitia Andrique, Carlos Ureña Martin. [11]These authors jointly supervised this work: Pierre Nassoy, Danijela Matic Vignjevic. pierre.nassoy@u-bordeaux.fr; danijela.vignjevic@curie.fr

fibroblasts in a hollow permeable shell made of alginate using the Cellular Capsule technology[16] (Fig. 1a). We monitored the dynamics of co-cultures over 15 days using bright-field and fluorescence time-lapse microscopy. As observed previously for co-culture of suspended cells[17–19] we anticipated different sorting or mixing scenarios after encapsulation: (i) the two cell types mix, resulting in a single spheroid with a salt and pepper-like pattern; (ii) cells completely segregate, making two separate spheroids; (iii) cells partially segregate with one cell type enwrapping the other (Fig. 1b). During the first 5 days of co-culture, when cells occupied only part of the total volume of the capsule, cancer cells, and fibroblasts self-sorted, forming two

spheroids composed exclusively of one cell type (Fig. 1c and Suppl. Movie 1). The contact area between spheroids started increasing after about 7 days of co-culture when spheroids deformed and filled the unoccupied space. Finally, after about 9 days, the 3D confluence was reached as spheroids filled up the capsule. The continued growth of spheroids after this stage caused dilation of the capsules as previously reported for encapsulated monocultures[16]. However, there was also a drastic tissue reorganization (Fig. 1c). After about 10 days, fibroblasts started relocating between the alginate shell and cancer cells, and, in about 24 h, they completely enwrapped the cancer cell spheroid (Fig. 1d, e).

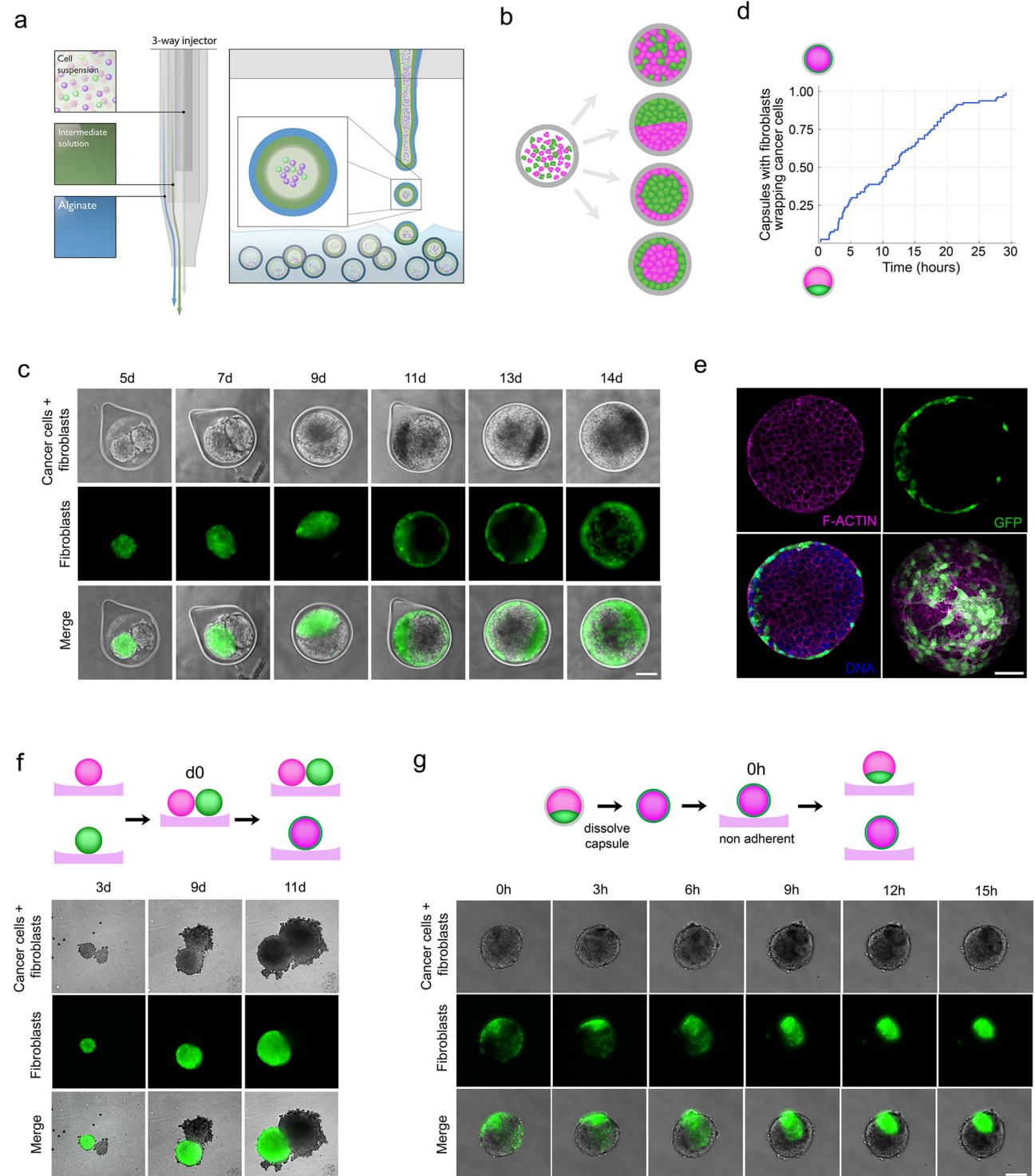

**Fig. 1 | Fibroblasts envelop cancer cells under confinement. a** Schematic representation of the encapsulation. Cells are encapsulated into hollow alginate spheres using a three-way injector consisting of the outermost phase containing alginate, the intermediate phase containing sorbitol and the innermost phase containing cells. Gelation of alginate occurs upon the fall of droplets into the gelation bath containing calcium chloride hexahydrate. Capsules were then filtered and transferred to the appropriate culture medium within less than 5 min. **b** Schematic representation of possible outcomes after encapsulation of cancer cells and fibroblasts as single cells into alginate capsules. **c** Evolution of co-culture of HT29 cancer cells and GFP expressing NIH3T3 fibroblasts over time in alginate capsules. Time $t = 0$ corresponds to the encapsulation of cells. Time represented in days, d First row: Phase contrast image showing cancer cell and fibroblast spheroids; second row: epifluorescent image of fibroblasts expressing GFP (green); last row: Merge. Scale bar,

100 μm. **d** Percentage of capsules in which fibroblasts envelop spheroids of cancer cells over time. $t = 0$ corresponds to the confluent stage. $n = 79$ capsules. **e** Confocal image of a fixed spheroid. All cells were visualized by staining F-actin (phalloidin, red) and DNA (DAPI, blue). Fibroblasts were discriminated as cells expressing GFP (green). Left, equatorial slices. Bottom right, maximal projection. Scale bar, 50 μm. **f** Evolution of co-culture of HT29 cancer cells and GFP expressing NIH3T3 fibroblasts over time on agarose-coated individual wells. Time $t = 0$ corresponds to seeding of cells. Time represented in days, d First row: Phase contrast image showing cancer cell and fibroblast spheroids; second row: epifluorescent image of fibroblasts expressing GFP (green); last row: Merge. Scale bar, 100 μm. **g** Evolution of cancer cells spheroids enwrapped with GFP expressing NIH3T3 fibroblasts over time after removal of alginate capsules. Time $t = 0$ corresponds to capsule dissolution, thus the release of the confinement. Time represented in hours, h Scale bar, 100 μm.

To find out if this phenomenon, which was observed in all the encapsulated NIH3T3/HT29 co-cultures that we investigated ($n = 79$), could be more general with other cell types, we performed additional experiments. First, we used non-immortalized CAFs isolated from the tumors of colon cancer patients and co-cultured them with HT29 cancer cells. Due to the much slower proliferation of primary CAFs compared with cancer cells, CAFs were scarcer in the capsules. Nevertheless, after confluence, similarly to NIH3T3 fibroblasts, primary CAFs spread over cancer cells and adopted a needle-like shape resembling that of CAFs observed in vivo (Suppl. Fig. 1a). Second, instead of fibroblasts, we used CT26 cancer cells that exhibit mesenchymal features, and we found that they also enwrapped HT29 cancer cells (Suppl. Fig. 1b, c), suggesting that mesenchymal cells enwrap more epithelial cancer cells under physical confinement. Finally, we have investigated whether this is a feature exclusive to mesenchymal and epithelial pairings. To this end, we encapsulated NIH3T3 cells (mesenchymal) with either CT26 cancer cells, which are fully mesenchymal, or HCT116 cancer cells, which exhibit an intermediate epithelial-mesenchymal phenotype. We observed that NIH3T3 and CT26 did not segregate post-encapsulation but remained intermixed, even following confinement (Suppl. Fig. 1d). Conversely, NIH3T3 and HCT116 cells segregated and primarily maintained this segregation post-confinement, with a few NIH3T3 cells intermixed with the cancer cells. We did not see NIH3T3 cells spreading over the surface of these cancer cells (Suppl. Fig. 1e).

Collectively, these experiments indicate that the typical organization of carcinoma in situ can only be achieved with a combination of fully epithelial and fully mesenchymal cells. This suggests that once cancer cells initiate the epithelial-to-mesenchymal transition, they begin to intermingle with fibroblasts, as observed in invasive carcinoma.

To further test whether confinement was required to induce the specific organization of cells, we co-cultured NIH3T3 fibroblasts and HT29 cancer cells in non-adhesive agarose wells, i.e., in the absence of confinement. As within the capsule, cells segregated and formed individual spheroids. However, even though both types of spheroids were kept in proximity over 15 days, we never observed fibroblasts enveloping cancer cells spheroids (Fig. 1f). This shows that spatial confinement is required to induce the reorganization of the fibroblasts around cancer cells.

Finally, we tested whether confinement is required to maintain this cellular organization. We dissolved the capsule once fibroblasts had enwrapped the cancer cell spheroid. Within 10 h upon release of the confinement, fibroblasts regrouped into a homogeneous cluster attached to the cancer cell spheroid taking a configuration identical to the initial stage of co-culture and resembling a dewetting process (Fig. 1g and Suppl. Movie 2). Dewetting, which is widely studied in soft matter physics[20], generally corresponds to the process of retraction of a fluid from a surface and was already reported for cell monolayers on solid surfaces[21,22].

Altogether, these data show that co-cultured cancer cells and fibroblasts can exhibit a self-organization transition into structures with fibroblasts enwrapping aggregates of cancer cells. This transition only occurs upon spatial confinement and is reversed upon confinement release.

## Cell–cell and cell–matrix adhesions are required for fibroblast spreading

Confinement of cells in 2D using a "cell confiner" was shown to trigger cell migration[23]. Thus, we hypothesize that fibroblasts spreading in our system could be due to increased cell migration speed under confinement. To get insight into cell dynamics, we performed two-photon live imaging at the onset of confluence (Fig. 2a and Suppl. Movies 3, 4). To quantify fibroblasts' migration before and after confluence, we tracked the trajectories of individual fibroblasts at the spheroid periphery and extracted their instantaneous speed and persistence path (Fig. 2b, c). We found that the directional persistence of fibroblasts, defined as the ratio of displacement to trajectory length, increased from 0.28 before confluence to 0.42 after confluence ($n = 18$ (before), $n = 23$ cells (after)), which is expected from geometrical constraints since the enwrapping fibroblasts were confined between the alginate shell and the cancer cell spheroid (Fig. 2c). However, by contrast with previous reports of increased migration speed for mechanically stressed cancer cells in 3D spheroids[24] we found that confinement does not affect the instantaneous speed distribution of fibroblasts within these encapsulated co-cultures (Fig. 2b). The initiation of spreading was characterized by fibroblasts forming chains of cells that migrated collectively over cancer cells. Only on a few occasions fibroblasts detached from the chain and migrated as individual cells. This phenotype is strikingly different from the one of confined amoeboid cells[23] and is rather reminiscent of the moving cell network of neural crest cells[25] and other mesenchymal cells[26].

Cell–cell and cell–matrix adhesions are important regulators of collective cell motility, which could cause fibroblasts to reorganize[27–29]. To evaluate the impact of cell–cell adhesions on fibroblast spreading, we stably depleted N-cadherin in fibroblasts (Fig. 3a) and encapsulated them with HT29 cancer cells. We found that in contrast to control fibroblasts that enwrapped cancer cells one day after confluence, which is now taken as reference time, the spreading of N-cadherin-depleted fibroblasts was delayed and by day 4, only about 60% of capsules had fibroblasts enwrapping cancer cells (Fig. 3b and Suppl. Fig. 4). Subsequently, we examined the potential impact of cell–matrix adhesions on the spreading process. Fibroblasts are the main producers of the extracellular matrix (ECM), in particular, fibronectin. To investigate the potential role of fibronectin in fibroblast spreading, we used five distinct siRNA probes to silence fibronectin expression in fibroblasts, each resulting in varying degrees of fibronectin depletion (Fig. 3c). Full depletion of fibronectin (probe #3) almost completely abolished spreading of fibroblasts, and 4 days after confluence only in less than 5% of the capsules, fibroblasts enwrapped cancer cells (Fig. 3d). Partial depletion of fibronectin (probe #2), however, only delayed the spreading of fibroblasts, and after 3 days spreading was completed in about 80% of capsules (Fig. 3d). This suggests that to spread, fibroblasts need to lay out the adhesive substrate. We thus stained our co-cultures for fibronectin prior to and after confluence. At the early stage, before confluence, the fibroblast spheroids were enriched in dense fibrillar bundles of fibronectin mainly localized between cells, suggesting that fibronectin mediates cell aggregation in fibroblasts (Fig. 3e). At the onset of fibroblasts spreading over cancer cells, thick bundles of fibronectin were still observed in between cells in the core of fibroblast aggregates (Fig. 3f, region 1),

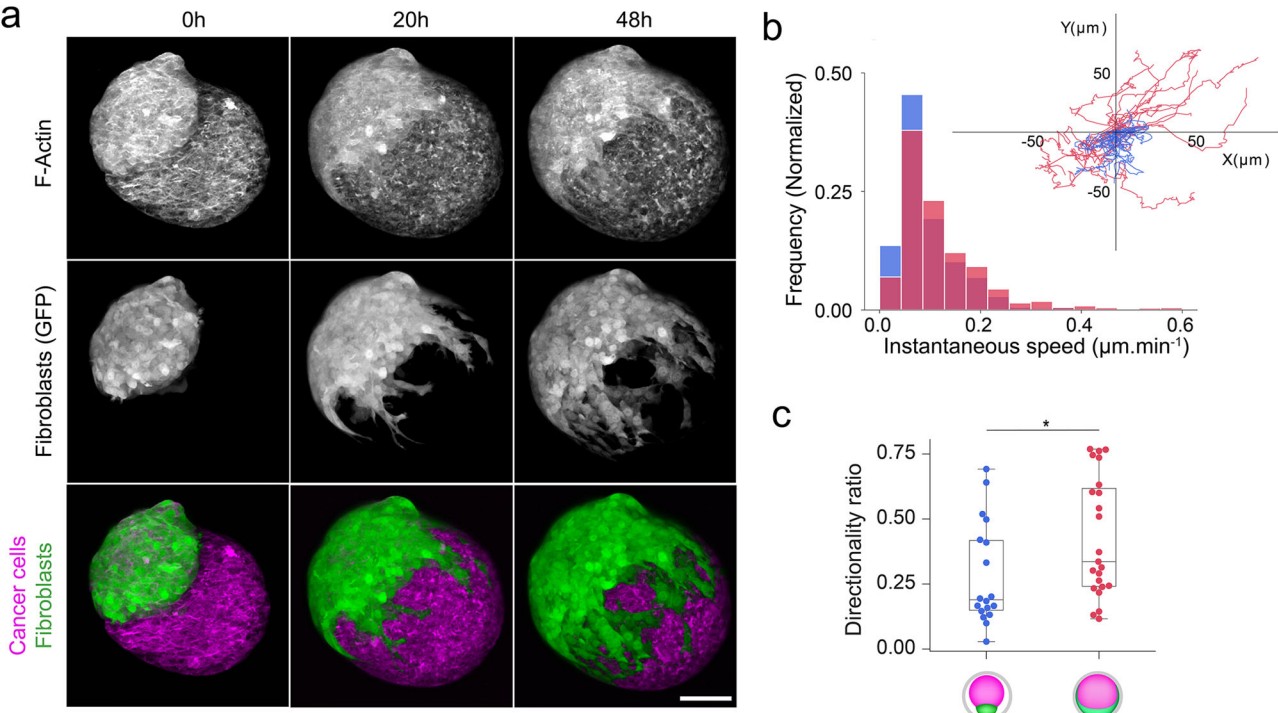

**Fig. 2 | Confinement does not increase cell migration of fibroblasts. a** Two-photon live imaging of a co-culture of cancer cells stained by membrane die FM 4-64 (red) and fibroblasts expressing GFP (green) at the onset of confluency. Time in hours, h. Scale bar: 50 μm. **b** Distribution of the instantaneous speed prior (blue bars) and after confluence (red bars). Reconstituted fibroblast 2D trajectories prior to confluence (blue line) and after confluence (red line). Time step: 20 min. Typical trajectory length: 15 h. **c** Path persistence prior to confluence (blue) and after confluence (red). Wilcoxon rank-sum test was performed (*$p < 0.05$). $n = 38$ cells.

however, at the front of spreading cells, fibronectin was detected only below the cells at the interface with the capsule (Fig. 3f). The appearance of the network also changed and consisted of thinner and more sparse fibers. To our surprise, we have not detected fibronectin at the cancer cell surface, suggesting that the fibroblasts spreading is not due to an increase in the affinity between cancer cells and fibroblasts. At the final stages, when fibroblasts completely enwrapped cancer cells, we observed that the fibronectin network was mainly localized at the interface with the capsule as a continuous 2D layer (Fig. 3e).

Altogether, these findings highlight the necessity of both cell–cell and cell–matrix adhesions for the chain-like migration of fibroblasts over cancer cells. They further suggest that while fibronectin may favor aggregation of fibroblasts before spreading, it allows their spreading by providing an adhesive substrate over alginate.

**The difference in surface tension cannot explain fibroblast organization over cancer cells**

The initial segregation process followed by confinement-induced spreading of fibroblasts over cancer cells phenotypically resembles developmental processes during which tissues separate and envelop one another. For example, germ layer segregation or epiboly during zebrafish gastrulation are tissue segregation and spreading processes, respectively, that are mediated by changes in tissue mechanical properties[30–32].

The interplay of surface tensions and interfacial tension of the two cell types can lead to the tissue to organize into distinct morphologies, including (i) a side-by-side morphology and (ii) a core-shell morphology. Scenario (i) corresponds to the unstressed configuration observed within the capsules before the confluence. Scenario (ii) yields the final configuration observed in the late stages after confluence. Upon confinement, we observed a gradual, continuous transition from morphology (i) to (ii).

To gain insight into the mechanism for the observed morphological transition under confinement, we measured the surface tension of fibroblasts and cancer cell spheroids using micropipette aspiration[33,34] (Fig. 4a).

These experiments were performed on monocultures of cancer cells and fibroblasts in the absence of confinement (i.e., freely growing spheroids made of individual cell types in the absence of encapsulation) and 3 days after confluence, corresponding to the stressed configuration. For the compressed spheroids, the capsule was dissolved, and pipette experiments were carried out within the next 30 min to ensure maintenance of the core-shell configuration and minimal change of all components that may affect surface tension (Fig. 1g). We found that the surface tension of fibroblasts spheroids, $\gamma_f^0$, is about three times higher than the surface tension of cancer cells spheroids $\gamma_c^0$ (Fig. 4b). These measurements were confirmed using the microplate compression technique (Fig. 4c–e). Despite differences in the absolute values of surface tensions, both techniques yielded values that highlight the same relative trend. The microplate compression technique, which requires analyzing the spheroid shape and force relaxation, could only be used for spheroids composed of either cancer cells or fibroblasts. Thus, to analyze surface tension in mixed spheroids, we only used micropipette aspiration. From micropipette aspiration experiments on spheroids just released from confinement, we measure a twofold increase of the surface tension $\gamma_c^c$ of cancer tissue, while the surface tension of fibroblast spheroids $\gamma_f^c$ is hardly altered by compression (Fig. 4b). This reveals that after confinement the mechanical properties of the cancer cell aggregates were altered. To discern whether the elevated surface tension in cancer cell aggregates could stimulate fibroblast spreading, we sought methods to reduce the surface tension of these aggregates. The most intuitive approach involves inhibiting myosin II, given its role in actomyosin tension, which constricts the cellular aggregate into a shape with minimal surface area[35]. Thus, we treated cancer cells and fibroblast spheroids with blebbistatin, an inhibitor of myosin II. As expected, this treatment lowered the surface tension of both, cancer cell's and fibroblast's, aggregates (Suppl. Fig. 2a). However, upon treating co-cultured capsules with blebbistatin or Y-27632, a Rho-associated protein kinase (ROCK) inhibitor, we didn't observe fibroblast spreading before reaching confluence, nor did we note any hindrance to fibroblast spreading post-confluence (Suppl. Fig. 2b–d). As this treatment

**Fig. 3 | Cell–cell and cell–matrix adhesions are required for fibroblast spreading. a** Western blot showing N-cadherin expression level in Control fibroblasts (Ctrl, transfected with scrambled shRNA) and fibroblast depleted from N-cadherin (shN-cadh). GAPDH is used as a loading control. **b** Frequency of capsules in which fibroblasts enveloped cancer cells. Capsules contain cancer cells with control or N-cadherin-depleted fibroblasts. $t = 0$ corresponds to the confluent stage. n ≥ 30 capsules per condition. **c**. Western blot showing Fibronectin expression level in Control fibroblasts (shCtrl, transfected with scrambled shRNA) and fibroblast depleted from fibronectin (shFN) using five different shRNA probes. GAPDH is used as a loading control. **d** Frequency of capsules in which fibroblasts enveloped cancer cells. Capsules contain fibroblasts showing different degrees of fibronectin depletion. $t = 0$ corresponds to the confluent stage. $n = 40$ capsules. **e** Confocal images of co-culture at day 5, which corresponds to the early stage, before confluency (two upper rows) and 10 days, which corresponds to the final stage with fully spread fibroblasts (two bottom rows). Both cell types are labeled with phalloidin (F-actin, red), fibroblasts express GFP (green), fibronectin is labeled with antibodies (magenta). Scale bar: 100 μm. **f** Confocal images of co-culture at the onset of fibroblasts spreading. Fibroblasts expressed GFP (green), cancer cells are unstained, and fibronectin is labeled with antibodies (magenta). Insets, higher magnification of boxed regions. Scale bars: 20 μm.

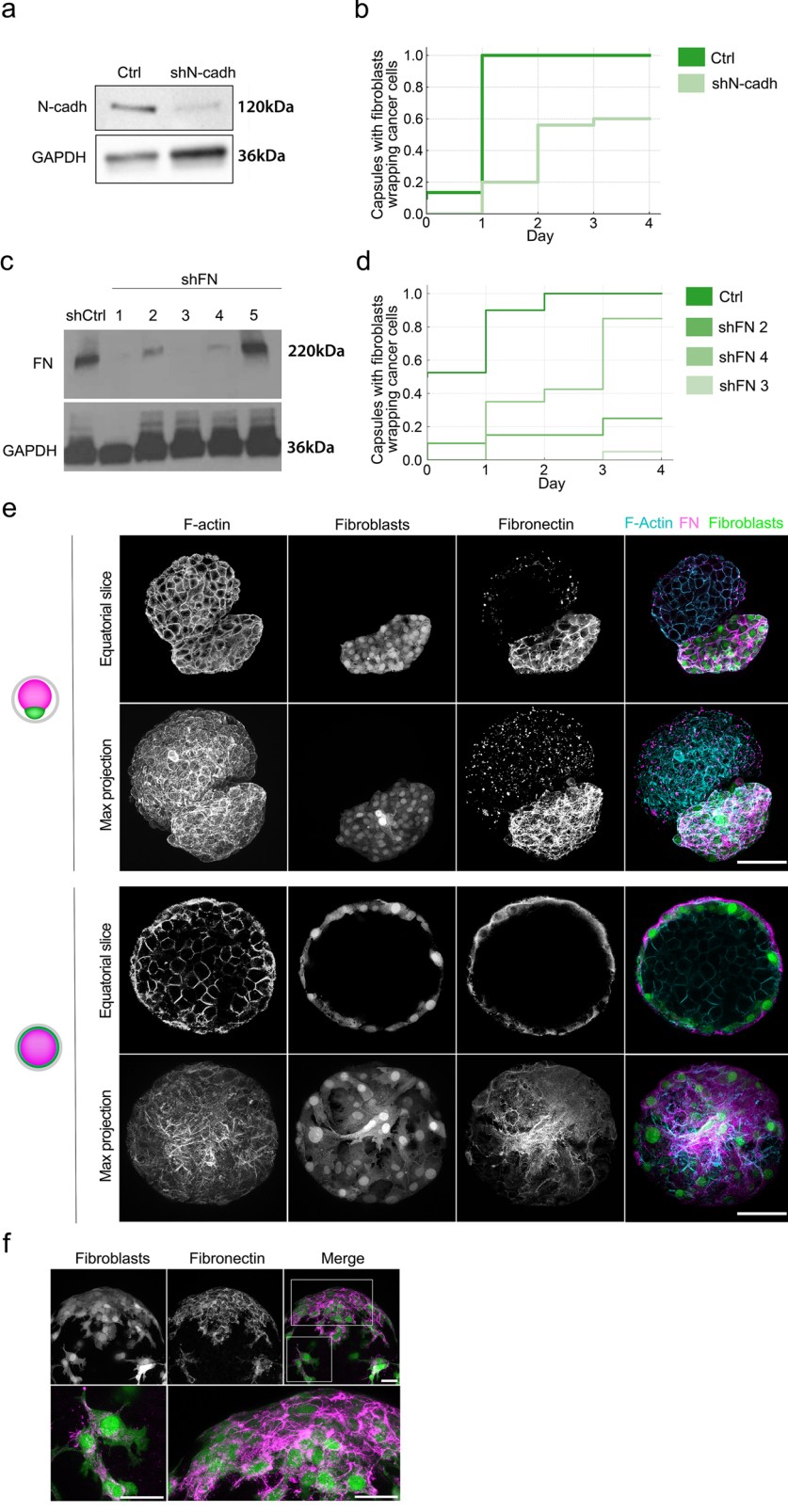

affects contractility, and thus surface tension, in both cell types, we next depleted myosin IIA specifically within cancer cells (Fig. 4f), leading to an anticipated reduction in the surface tension of cancer cell aggregates (Fig. 4g). We subsequently encapsulated these myosin IIA-depleted cancer cells with wild-type fibroblasts, revealing a significant delay in fibroblast spreading. After 3 days, only 30% of capsules exhibited dispersed fibroblasts (Fig. 4h). Together, these results suggest that an increase in surface tension of cancer cell aggregates is necessary to drive fibroblast spreading.

We further assessed theoretically whether this change in surface tension could drive the transition. The model relies on the hypothesis that

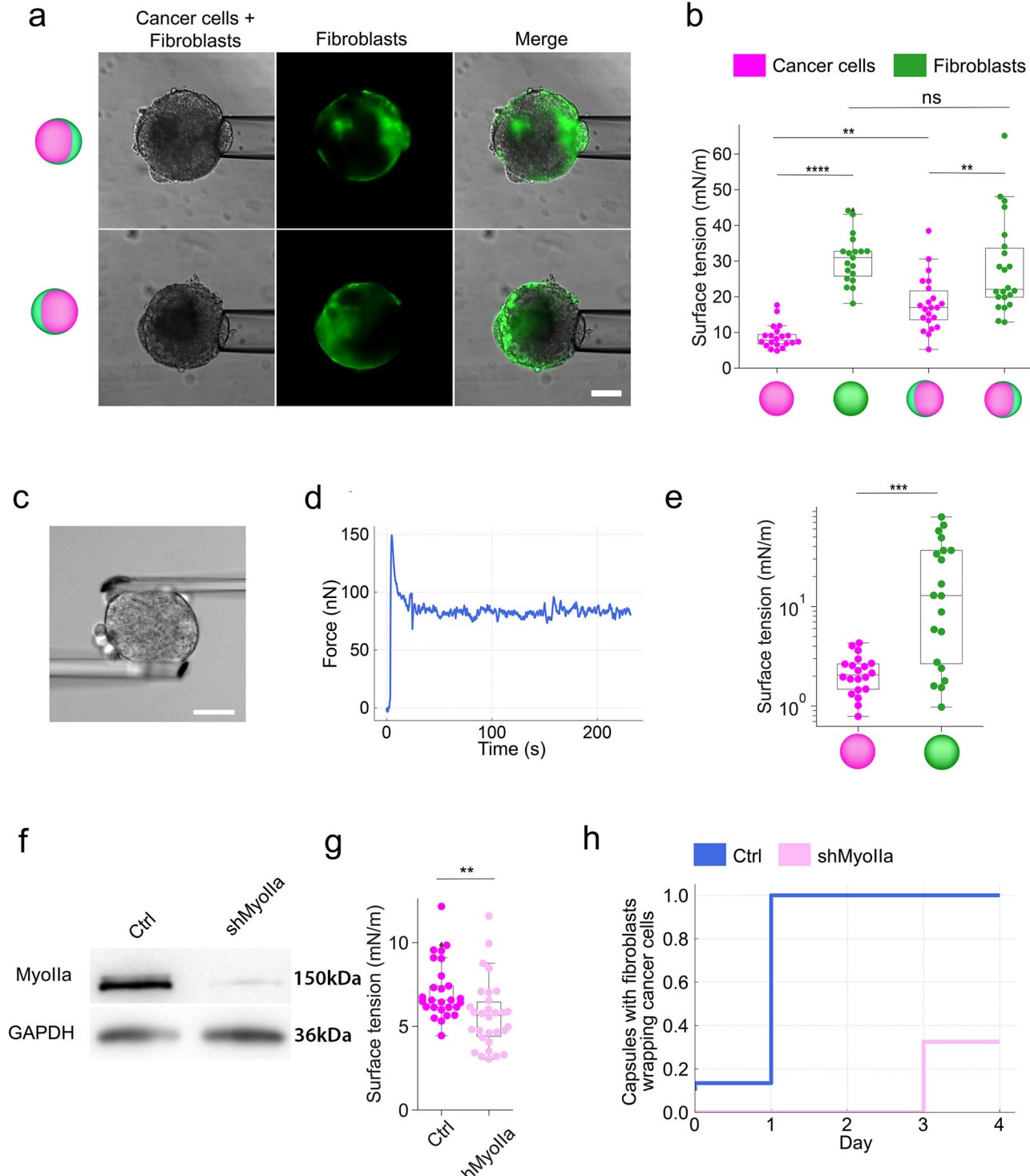

**Fig. 4 | The surface tension of fibroblast spheroid is higher than cancer cells.**
**a** Measurement of surface tension using the micropipette aspiration assay. First row: Phase contrast and epifluorescent image showing micropipette aspiration of HT29 cancer cell side; second row: Phase contrast and epifluorescent image showing micropipette aspiration of GFP expressing NIH3T3 fibroblasts side. Scale bar, 50 μm. **b** Individual (dots) surface tension measurements on cancer cell spheroids (red) and fibroblast spheroids (green) growing without confinement and co-cultures after the onset of fibroblast spreading. $N = 3$ runs of encapsulation, $n = 82$ capsules. One-way ANOVA test was performed (***$p < 0.001$). Three independent experiments were performed. $n = 82$ capsules. **c** Measurement of a spheroid using microplates assay. Scale bars: 25 μm. **d** Evolution of force as a function of time in a microplate assay. **e** Individual (dots) surface tension measurements on cancer cell spheroids (red) and fibroblast spheroids (green) growing without confinement. One-way ANOVA test (***$p < 0.001$). $n = 40$ spheroids from three independent experiments. **f** Western blot showing myosin IIA expression level in control HT29 cancer cells (Ctrl, transfected with scrambled shRNA) and myosin II-depleted cancer cells (shMyosinIIa). GAPDH is used as a loading control. **g** Individual (dots) surface tension measurements on cancer cell spheroids made of control and myosin IIa-depleted cancer cells growing without confinement. One-way ANOVA test was performed (**$p < 0.01$). Three independent experiments were performed. $n = 59$ capsules, from $N = 3$ experiments. **h** Frequency of capsules in which fibroblasts enveloped cancer cells. Capsules contain fibroblasts and control or myosin IIa-depleted cancer cells. $t = 0$ corresponds to the confluent stage. $n \geq 40$ capsules per condition.

https://doi.org/10.1038/s42003-024-05883-6 **Article**

spheroids behave as liquids, which is first supported by the observation of a dewetting process upon confinement release (Fig. 1g and Suppl. Movie 2). Further, we checked that an alternative view where cells behave as elastic solids allowing soft fibroblasts to squeeze in between the alginate wall and the spheroid of stiff cancer cells does not hold. Indeed, the measure of Young's modulus Y of fibroblasts and cancer cells within mono-cultured spheroids using the microplate technique (see Methods section) shows that fibroblasts are slightly stiffer than cancer cells (Suppl. Fig. 3a). Thus, by treating spheroids as liquids, we propose a surface energy model that may explain how the tissue phase separates and organizes into distinct morphologies, followed by a mechanism for the transition from one morphology to the other. By examining the interplay of surface tensions and interfacial tension of the two cell types in the absence of confinement, this model predicts that: (i) a side-by-side morphology in which contact area between the two spheroids, one made of fibroblasts and the other of cancer cells, is favored when the interfacial tension is large compared to the difference in surface tensions of the two spheroids, while (ii) a core-shell morphology, with a very large contact area between the cells forming the core and the ones forming the shell, will be expected when the interfacial tension and the difference in surface tensions are comparable, with the spheroid with the lower surface energy forming the core (Details in SI). In Suppl. Fig. 3b (left), we show the conditions that will favor a side-by-side morphology versus a core-shell morphology where the fibroblasts enwrap the cancer cells.

Once the system is confined as in the experiments, our calculations suggest that it can attain a lower energy with a Janus morphology (i.e., where both spheroids form spheres with extended interfacial contact) compared to a side-by-side morphology with minimal contact, for surface and interfacial tensions (Suppl. Fig. 3b, right) (see Suppl. Notes 1–3 for details). Next, we explain a potential mechanism for the transition from the experimentally observed Janus configuration, which is a constrained configuration, to a core-shell configuration. To contextualize the surface energy measurements and the observed morphologies, we introduce the contact angle θ of the system when it is confined within the capsule and explore geometry as it is changed. The contact angle is related to the surface energies through Young's equation, $\cos\theta = (\gamma_f^c - \gamma_c^c)/\gamma_{cf}$, where θ is measured within the cancer cell phase (Suppl. Fig. 5 and Suppl. Note 2), and $\gamma_{cf}$ is the interfacial energy between cancer and fibroblast spheroids. This relation states that as Δγ decreases, the contact angle should increase, which means that the system is moving towards a state where the fibroblasts start to preferentially "wet" the capsule walls (θ > 90°). In Suppl. Note 2, we relate more precisely the geometry of the confined Janus configuration to the surface energies and volume fraction, illustrating this picture. In summary, theoretical analysis proposes that confinement could lower the energy barrier impeding the fibroblast wrapping of cancer cells (Suppl. Notes 1–3 and Suppl. Figs. 5–7). One approach to lowering this energy barrier involves reducing the surface tension of the fibroblast aggregate. To achieve this, we depleted myosin IIA specifically in fibroblasts (Suppl. Fig. 3c). Although this depletion led to only a marginal decrease in surface tension (Suppl. Fig. 3d), potentially due to the compensatory actions of other myosin isoforms[36], it did result in a significant impairment of fibroblast spreading over control cancer cells (Suppl. Fig. 3e). Likewise, depletion of N-cadherin (Suppl. Fig. 3f) or fibronectin (Suppl. Fig. 3g) significantly reduced surface tension as anticipated[37] yet the fibroblasts remained unable to spread (Fig. 3b, d).

Collectively, our results indicate that despite the convergence of surface tensions between cancer cell and fibroblast aggregates upon confinement, the fibroblasts' surface tension remains higher. Consequently, the differential surface tension theory cannot simply explain the spatial arrangement of fibroblasts around cancer cells under confinement.

**The buildup of compressive stress upon confinement is required to induce fibroblasts reorganization**

Finally, we wondered whether confinement alone (i.e., spatial constraints or cell crowding in a closed volume) would be sufficient to drive fibroblasts reorganization or if this process would result from the associated mechanical stress (e.g., a threshold in the compressive force applied to the spheroids). As previously done[16], we took advantage of the elastic properties of the alginate capsule to compute the mechanical stress exerted by the growing aggregate. The pressure exerted by the growing aggregate of cells onto the elastic capsule, or conversely, the restoring pressure exerted by the capsule onto the aggregate, can be derived from capsule deformation according to a variant of Hooke's law for a spherical hollow spring:

$P = \frac{2E}{1-\nu} \frac{h}{R} \frac{u(R)}{R}$ where $u(R)$ is the radial displacement at a distance $R_{in} \leq R \leq R_{out}$ from the center of the capsule, E the elastic modulus of the alginate gel (E = 68 kPa)[16], h the thickness of the capsule, $\nu$ the Poisson's ratio ($\nu = 1/2$). We thus monitored the pressure exerted on the capsule as a function of time (Fig. 5a, b). Compared with previously studied mono-cultures of mesenchymal-like cancer cells[16], the magnitude of the pressure here was about twofold larger, but the overall evolution was similar. From the representative curve shown in Fig. 5c, we observe that pressure rapidly increased within the first ~30 h at a rate of 0.55 kPa/h, followed by an abrupt decrease in the rate of pressure buildup (~0.22 kPa/h). Then, by measuring the pressure exerted on the microtissue at the onset of spreading over 79 capsules (Fig. 5d), we observed that, on average, fibroblast enwrapping occurred at a large pressure P = 5 ± 3 kPa (mean ± SD). Despite significant variability in spreading pressure, it remains noteworthy that most fibroblasts start to spread only when the pressure reaches at least 2 kPa (Fig. 5d). This clearly indicates that spatial confinement, without pressure buildup, may be insufficient to induce fibroblast spreading, suggesting a mechanosensitive mechanism.

To reduce pressure buildup, we blocked proliferation and compared the outcome with continuously growing spheroids. We incubated the capsules with an anti-proliferative drug, mitomycin C, at time T = 0 corresponding to confluence (Fig. 5e). Upon addition of mitomycin C, spheroids continued increasing in size but at a slower rate than control spheroids, and they were ultimately arrested about 15 h after confluence (Fig. 5e). In control groups, the number of capsules in which fibroblasts spread around cancer cells increased over time as pressure built up, reaching ~90% of capsules with spread fibroblasts at 35 h (Fig. 5e). For mitomycin C-treated co-cultures, the percentage of capsules with spread fibroblasts at a given time point was always lower than for the control capsules, which is consistent with the fact that pressure, was reduced. Remarkably, in the time window over which growth was fully inhibited (typically 15–35 h), the percentage of capsules with fibroblast spreading remained roughly constant at about 25%. Thus, maintaining the spheroids under low pressure for an extended period was insufficient to drive further fibroblast spreading.

Together, these data show that pressure buildup, and not geometrical confinement alone, is necessary to induce fibroblasts to spread over cancer cells.

## Discussion

In this study, we have developed a 3D in vitro model that recapitulates an organization reminiscent of CAFs encapsulating tumor cell clusters as observed in vivo. We used the cellular capsule technology to co-culture cancer cells and fibroblasts. We found that a buildup of compressive stress, resulting from geometrical confinement and spheroid volume increase, is essential to initiate and maintain the arrangement of fibroblasts around cancer cells. This compressive stress correlates with enhanced surface tension in the cancer tissue, while it doesn't seem to influence the mechanical properties or migration speed of fibroblasts. Interestingly, the compression prompts a profound reorganization of the fibroblast microenvironment, specifically coinciding with the restructuring of the fibronectin network during the spread of fibroblasts over cancer cells. In the absence of compression, fibronectin is positioned between interacting fibroblasts, potentially serving a "glue-like" function akin to that observed in primary fibroblasts[29]. A simple surface energy-based calculation shows that the region in phase space where fibroblast wrapping is favored is extended upon spatial confinement (Fig. S2A. More specifically, the onset of wrapping may occur with a higher probability when the difference between $\gamma_f$ and $\gamma_c$ is reduced or/and when the interfacial tension $\gamma_{fc}$ vanishes. On the one hand,

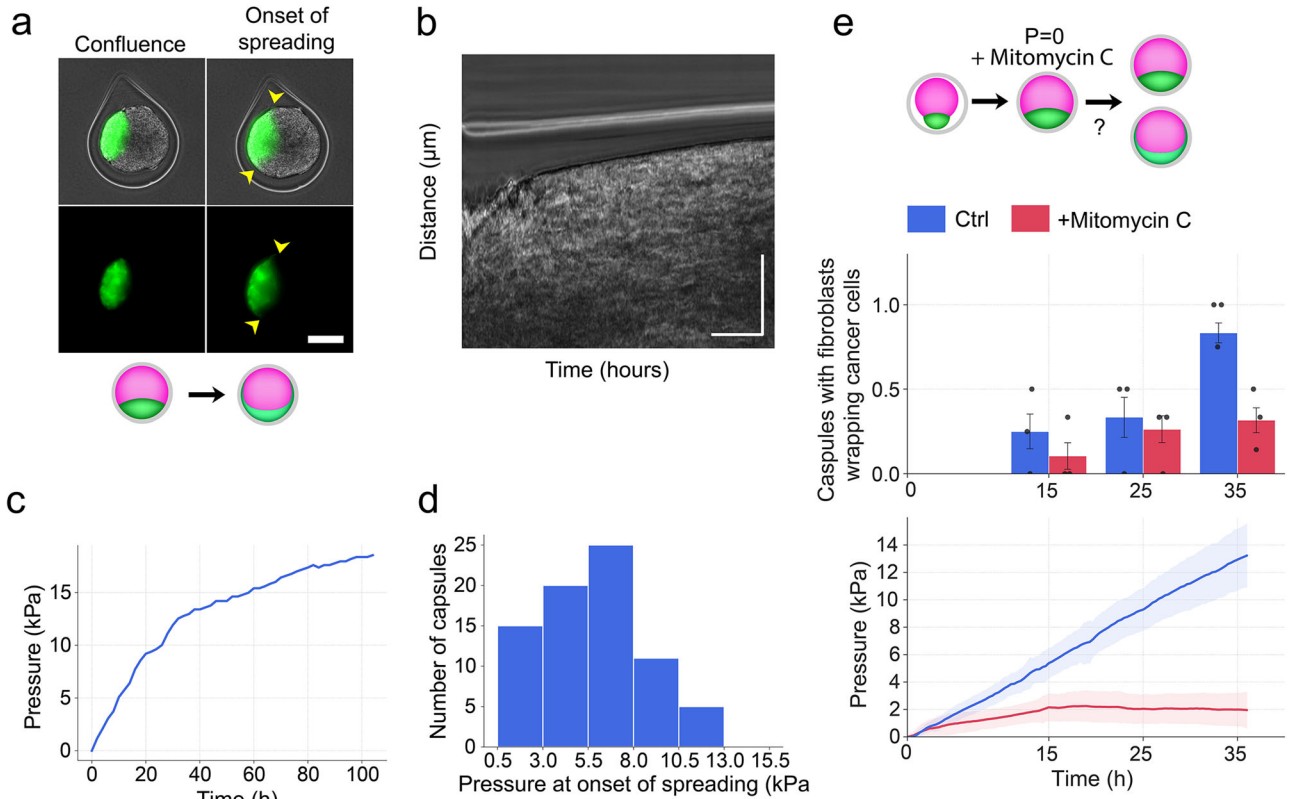

**Fig. 5 | The buildup of pressure triggers fibroblasts spreading. a** Images representing the onset of confluence and onset of spreading. The yellow arrowhead represents the early onset of spreading. **b** Phase-contrast intensity plot as a function of time and radial distance from the two spheroids center. Scale bars, 20 h and 50 µm. **c** Pressure evolution over time, starting from confluence ($t = 0$). **d** Frequency of capsules with specified pressure at the onset of fibroblast spreading. $n = 79$ capsules. **e** Lower graph, the volume of the capsule normalized to the volume at the onset of confluence as a function of time for control (blue line) and cells treated with mitomycin C (red line). Gray shadow: standard deviation. Time $t = 0$ corresponds to confluence. (Upper graph) Frequency of capsules in which fibroblasts started to envelop cancer cells at different time points (T1 = 15 h, T2 = 25 h, and T3 = 35 h) for control (blue bars) and mitomycin C-treated co-cultures (red bars). $n = 31$ capsules from $N = 3$ independent experiments. Values are mean ± standard error.

we are not aware of any experimental means to measure $\gamma_{fc}$. On the other hand, contrary to what is expected, decreasing the surface tension of the fibroblast aggregate by depleting fibronectin or N-cadherin did not expedite spreading. It even impeded it, implying that the cell–cell adhesions between fibroblasts and cell–matrix adhesions between fibroblasts and substrate are crucial for spreading, although the detailed molecular mechanism remains to be unraveled. This spreading process begins with fibroblast migration as chains of cells, presenting a phenotype noticeably distinct from the confined ameboid cells[23] and somewhat reminiscent of the migratory cell network of neural crest cells[25] and other mesenchymal cells[26]. In terms of physical properties, this digitated pattern also draws parallels with the Saffman–Taylor fingering observed between viscous fluids[38], as well as the Rayleigh–Taylor instability in elastic slabs or wedges[39,40].

We believe that this in vitro model constitutes a unique tool to investigate the crosstalk between CAFs, pressure, and the tumor at the pre-invasive stage of carcinoma in situ. For example, it could be used to study if CAFs stimulate or restrain tumor progression. By accumulating around the tumor, CAFs can induce stiffening of the ECM, causing accumulation of compressive stress[15], which was shown to slow down tumor growth in vitro[16,41,42]. On the other hand, the accumulation of compressive stress might also enhance tumor invasion[16,24]. Altogether, we believe this complex 3D in vitro model system could represent a more physiological system to mimic tumors.

## Methods
### Cell lines and primary cell cultures
We used human colon carcinoma HT29 cells (ATCC HTB-38; American Tissue Culture Collection), mouse colon carcinoma CT26 (ATCC HTB-38;

American Tissue Culture Collection), mouse fibroblasts NIH3T3 stably expressing GFP (AKR-214, Cell Biolabs), and human primary non-immortalized fibroblasts (CAFs) isolated from fresh colon cancer samples from patients treated at Institut Curie and Lariboisière Hospital, Paris (4). Written consent from the patients and approval of the local ethics committee "Comité de Protection des Patients" was obtained. All cells were maintained in DMEM (Invitrogen) supplemented with 10% (vol/vol) FBS (Invitrogen) in a humidified atmosphere containing 5% $CO_2$ at 37 °C, with the medium changed every 2 days.

### Encapsulation of cells into alginate hollow spheres
Encapsulation of cells was performed as described in (9). Briefly, the outermost phase (AL) was prepared by dissolving 2.5% wt/vol sodium alginate (Protanal LF200S; FMC) in water and by adding 0.5 mM SDS surfactant (VWR International). The solution was filtered with two inline glass filters of 1 µm and 0.22 µm (Pall Life Science) and stored at 4 °C. The intermediate phase (IS) was a 300 mM sorbitol (MERCK) solution. The innermost phase (CS) was obtained by detaching sub-confluent cells from the culture flask with 0.5% Trypsin-EDTA (Invitrogen). After washing with culture medium, cells were spun (300 g, 3 min, 20 °C), and resuspended in 300 mM sorbitol solution at an approximate concentration of $3 \times 10^6$ cells/ml.

The three fluid phases (cell suspension (CS), intermediate solution (IS), and (AL) were loaded into sterile Teflon tubing (1/16 OD x 0.40 ID, IDEX). The tubing was connected on one end to 10 mL TLL syringes (Hamilton), and on the other end into the appropriate inlets of the co-extrusion device, which is clamped vertically to a post inside a laminar flow hood. The syringes were mounted on syringe pumps (Low Pressure Syringe Pump neMESYS, CETONI) that control fluid injection at the desired flow rates.

We mostly used one set of flow rates $q_{CS} = 20\ mL \cdot h^{-1}$, $q_{IS} = 20\ mL \cdot h^{-1}$, and $q_{AL} = 30\ mL \cdot h^{-1}$ to make "thick" capsules, i.e., typically with a shell width of 30 μm for a radius of 150 μm. After initiation of the flows, the compound microdroplets were directed to a gelation bath containing 100 mM calcium chloride hexahydrate (Merck) and traces of Tween 80 (MERCK), placed at ~0.5 m below the outlet of the device. Capsules were immediately filtered and transferred to the appropriate culture medium within less than 5 min. Cellular capsules were placed inside an incubator (37 °C, 5% $CO_2$). Cells aggregated to form spheroids within a few days.

After of several days of growth and formation of differentiated spheroids inside the capsules they were selected before reaching compression point, they were picked individually by pipetting them under a sterile cell culture hood equipped with a microscope. Size distribution aimed to be approximately on the 150 μm range.

### Immunofluorescence
Spheroids were fixed in 4% paraformaldehyde in PBS at RT for 40 min, which was sufficient to dissolve the shell. After washing with 0.01% BSA in PBS, the aggregates were incubated in 0.1% Triton X-100 in PBS (MERCK) for 40 min at RT. BSA was used to prevent spheroids from binding to the plastic of bottom dishes and pipette tip. After washing twice with 0.01% BSA in PBS, spheroids were incubated with primary antibodies diluted in PBS containing 0.01% BSA overnight at 4 °C. Specifically, we used fibronectin and vimentin (1:50, MERCK). After washing twice with 0.01% BSA in PBS, the spheroids were incubated in PBS containing 0.01% BSA and Alexa Fluor-conjugated secondary antibodies, Phalloidin Alexa Fluor 647, and DAPI (Thermo Fisher) at a ratio of 1:200, 1:200, and 1:500 respectively for 2 h at RT. The cell aggregates were finally washed four times in 0.01% BSA in PBS before mounting for imaging.

### Imaging of fixed and live spheroids
To prevent displacement or drift of the capsules during imaging, we designed custom-made holders. Holes (diameter of ~2 mm) were drilled in a 50 mm plastic-bottom Petri dish. A 20 mm square glass coverslip (0.16 mm thick) was glued to the bottom of the Petri dish using epoxy resin (Loctite 3430; Radisopares-RS Components). To prevent displacement of capsules, a 0.2% ultra-pure low-melting-point agarose (MERCK) solution made in a serum-free culture medium was prepared and cooled down at 37 °C. The percentage of agarose was chosen to generate minimal stress on the growing spheroids (~0.2%). Each well was filled with 4–5 spheroids or capsules mixed with 10–20 μl of the 0.2% agarose solution. After 10 min gelation of the agarose at RT, 10% FBS and 1% antibiotic-antimycotic in culture medium (for live imaging) or PBS (for fixed imaging) was added to each dish.

Spheroids growth inside the capsules was monitored by phase-contrast microscopy. Around 64 capsules were selected from the whole batch of cellular capsules and individually transferred to each well of homemade holders. Each capsule was imaged every 20 or 30 min up to 3–5 days using a Nikon Eclipse Ti inverted microscope (10×/0.3-N.A. dry objective; Nikon Instruments) equipped with a motorized stage (Märzhäuser) and climate control system (The Brick; Life Imaging Systems). The microscope and camera (CoolSNAP HQ2; Photometrics) were driven by Metamorph software (Molecular Devices). The microscope was equipped with a fluorescent lamp to capture the dynamics of NIH3T3 expressing GFP spheroid. The culture medium was renewed by one-half every 2 days. Imaged were processed using Fiji.

3D imaging was performed using an inverted Acousto Optical Beam Splitter two-photon, laser-scanning confocal microscope SP8 (Leica) coupled to femtosecond laser Chameleon Vision II (Coherent Inc) equipped with a 40×/0.95-N.A. oil-immersion objective. The microscope is further equipped with three non-descanned HyD (Hybrid) detectors: NDD1 (500–550 nm), NDD2 (≥590 nm) and NDD3 (450 nm). To monitor cell dynamics of HT29 cells inside the spheroid, we incubated the spheroid in FM 4-64 (Thermo Fisher) at a concentration of 2 μg/mL. Images were collected every 30 min for 42 h. Images were processed using Fiji and Imaris.

### Drugs essays
To block cell proliferation, spheroids were incubated in a culture medium supplemented with 20 ug/mL Mitomycine-C (Roche) for 4 h. The migration of cells was not impacted under this treatment. To inhibit cell contractility, spheroids were incubated in a culture medium using 50 μM myosin II inhibitor, blebbistatin (MERCK), or 100 μM ROCK inhibitor, Y-27632 (MERCK).

### siRNA and shRNA knockdowns
HT29 cells and NIH3T3 were infected with MyH9 shRNA lentiviral particles (aYAP, Gift), control shRNA Lentiviral Particles (Santa Cruz Biotechnology, sc-108080), N-cadherin shRNA lentiviral particles (Santa Cruz Biotechnology sc-35999-V) and Fibronectin shRNA lentiviral particles (SHCLNG, MERCK) and shCtrl (SHC002, MERCK). Cells were incubated at 37 °C with lentivirus in DMEM complete and Polybrene solution (8 μg.ml) and harvested by trypsinization 3 days after infection. Single-cell suspensions were sorted by flow cytometry according to high levels of reporter gene expression.

NIH3T3 cells were seeded into six well-plates (TPP, Trasadingen, Switzerland; 92106) and next day, transfection was performed with 40 nM small interfering RNAs (siRNA) using HiPerFect reagent (Qiagen, 301707) in Opti-MEM medium (Thermo Fisher Scientific, Courtaboeuf, France; 31985070), according to the manufacturer's instructions. After 3 days, cells were collected in the morning to follow a second transfection 8 h after seeding. Cells were washed after overnight incubation. The siRNAs used were as follows: Allstars negative control (Qiagen, Courtaboeuf, France; SI03650318) and Mouse ON-TARGETplus Mouse Myh9 (Horizon, LQ-040013-00-0005).

### Western blot
Cell lysates were obtained using a RIPA buffer solution (ThermoScientific, 89901) supplemented with a cocktail of protease (ThermoScientific, 78410) and phosphatase (ThermoScientific, 78420) inhibitors and kept at 4 °C. Protein concentration was calculated using the BCA Protein Assay kit (ThermoScientific, 23227). Then, a 4X Laemmli solution was added before blotting to obtain a final concentration of 50 mM Tris pH 6.8, 2% sodium dodecyl sulfate, 5% glycerol, 2 mM 1,4-dithio-dl-threitol (DTT), 2.5 mM ethylenediaminetetraacetic acid, and 2.5 mM ethylene glycol tetraacetic acid.

Proteins were fractionated by SDS-PAGE (4–20% TGX gels, BioRad, Marnes la Coquette, France; 456-8053) and blotted onto nitrocellulose membranes (BioRad, Marnes la Coquette, France; 1704159). The membranes were blocked with 5% BSA in PBS containing 0.1% Tween 20 (PBS-T) and hybridized with the primary antibody of interest overnight at 4 °C. Then, the membranes were washed in PBS-T and hybridized with the secondary antibody for 1 h at room temperature. Antibodies were diluted in PBS-T containing 5% BSA. After washes, immune complexes were revealed by chemiluminescence (Thermo Fisher Scientific, Courtaboeuf, France; 34580), imaged using the ChemiDoc™ Imaging Systems (BioRad, Marnes la Coquette, France), and analyzed with ImageJ.

### Analysis of spheroid growth and pressure
The average radius of a spheroid at each time point was defined by $R = \sqrt{\frac{S}{\pi}}$, where S is the equatorial cross-section of the spheroid. The equatorial cross-section of the spheroid was measured using a custom-made macro in Fiji. For each time point, the macro makes binary images of the spheroid from phase-contrast images. The confluence time ($t = 0$) was determined as the time for which spheroid growth exhibits an inflection point. We verified that this time coincides with the visual determination of confluence. For thick capsule, the pressure may be determined by considering the capsules as thin-walled pressurized vessels in the framework of isotropic linear elasticity[27]: $P = \frac{2E}{1-\nu}\frac{h}{R}\frac{u(R)}{R}$, where E is Young's modulus of alginate shell measured to be $E = 68 \pm 21$ kPa, $\nu$ is the Poisson's modulus ($\nu = 0.5$ according to volume conservation of the shell), h is the thickness of the alginate shell, $u(R) = \frac{(R(t)-R_0)}{R_0}$ is the radial displacement at a distance R from the center of the

capsule. To determine u(R), we monitored the evolution of R(t) following the protocol described above.

## Decapsulation assay

Using capsules in which NIH3T3 were spread around HT29 (11 days after confluence), alginate shells were dissolved by incubation in PBS 1X for 5 min at RT. Bare spheroids were then individually transferred in 96 wells plates coated with agarose cushion (1% in serum-free culture medium) and imaged overnight by phase contrast and epi-fluorescence imaging.

## Cells tracking and analysis inside spheroid

NIH3T3 trajectories inside the spheroid were tracked manually. The (x,y) position of the centroid of the nucleus was manually determined for each time point. For each cell, trajectories were reconstituted, concatenating positions of all centroids. Trajectories were stopped when the cell disappeared from the imaging plane or when the cell divided. Using a custom-made MATLAB (MathWorks) program, we computed various parameters of the trajectories: instantaneous speed and persistence.

The instantaneous speed is the speed of a cell at a specific time point t and is defined by $v(t) = \frac{\sqrt{(x(ti+1)-x(ti))^2 + (y(ti+1)-y(ti))^2}}{\Delta t}$.

## Micropipette experiments

Capsules containing spheroids made of a mix of cancer cells and fibroblasts were decapsulated about 3 days after confluence- once fibroblast started enwrapping cancer cells. Spheroids were placed in suspension in a non-adhesive glass-bottom culture dish and incubated at 37 °C and 5% $CO_2$ for microaspiration. As controls, we use spheroids made of single-cell types growing without confinement in non-adhesive agarose wells. The micro-aspiration setup[34] was built on an inverted Leica microscope equipped with an Eppendorf Transferman micro-manipulator holding micropipettes connected to a Fluigent MFCS EZ microfluidic pump. Images were acquired with a 40x/0.8NA dry objective. The surface tension at the cell-medium interface of spheroids was measured as previously described[34]. Surface tension was calculated using Laplace's law:

$$\gamma_{cm} = \frac{P_c}{2\left(\frac{1}{R_p} - \frac{1}{R_c}\right)}$$

in which $R_c$ is the resting radius of curvature of the spheroid at the location of the measurement, $P_c$ is the critical pressure at which spheroid deformation reaches $R_p$, the micropipette radius. Shape analysis was performed using ImageJ[43]. Of note, even though the time interval between shell dissolution and surface tension measurement, no longer than 30 min, was much shorter than the time required for fibroblasts to transit back to the sorted-out configuration (>6 h, Fig. 1f), we cannot rule out that the surface tension of aggregates was unchanged.

## Microplate experiments

A microplates-based rheometer[44] is used to measure the surface tension of cell aggregates. In such a device, the sample is compressed between two parallel microplates, one rigid, the other flexible, and of calibrated stiffness k. Thus, the force F applied to the sample is simply given by F = k δ, where δ is the flexible plate deflection. To determine the surface tension γ of the aggregates, the setup was used in "relaxation" protocol where the sample is submitted to a predefined deformation in compression and the plateau force $F_P$ to maintain such a deformation is measured. As reported in[45] γ can then be expressed as γ = $F_P/L_M$, where $L_M = \pi R_1[(R_1/R_2-1)]$ with $R_1$ and $R_2$ the radii of curvature of the aggregate at its median plane. Note that the microplate technique can conversely provide an estimate of Young's modulus of spheroids if one assumes a solid-like behavior. Young's modulus Y is derived from the measured force $F_p$, the contact area A between the plate and the spheroid, and the strain ε according to the equation: Y = F/(A.ε).

Since the set-up was initially designed to test single cells (~10 μm), the design of the flexible microplate had to be adapted to the size of the cell aggregates (~100 to 800 μm). Thus, new spatula-like plates have been designed with wide tips (1 mm in width) and a thin neck (around 5 mm long and 100 μm width) to be able to test the largest, still soft, cell aggregates. Moreover, since the force needed to compress the aggregates increases with the aggregate size, $F_p$ values ranged from ~50 nN to ~3 μN. Thus, we used three different flexible microplates of adapted stiffness -namely 11.4, 44, and 137 nN/μm- to cover this wide force range with a good resolution on the plate deflection δ. The flexible microplates were obtained by heating and pulling borosilicate plates of 10 cm×1 mm × 200 μm[46]. The stiffness of the flexible microplates was calibrated using a standard microplate itself calibrated following the protocol detailed in[44]). The setup was mounted on a Leica DMIRB inverted microscope (Leica Microsystems, Rueil-Malmaison, France), and samples were visualized using a x10 or a x20 objective, depending on the size of the aggregate. $R_1$ and $R_2$ were measured from bright field images of the compressed aggregates. $R_1$ was defined as $D_1/2$, where $D_1$ is the apparent diameter of the aggregate at mid-distance from the parallel plates. $R_2$ was obtained by fitting circles to the free edges of the aggregates.

## Statistics and reproducibility

All experiments were performed at least in triplicates, with the exception of Figs. 1f, g, 2a–c, 3b, 3d, 4h and Suppl. Figs. 2a–d, 3e that are from two to three replicates. Further information on research design is available in the Nature Reporting summary linked to this article.

## Reporting summary

Further information on research design is available in the Nature Portfolio Reporting Summary linked to this article.

## Data availability

All data needed to evaluate the conclusions in the paper are present in the paper and are available upon request. Uncropped western blots are available in Suppl. Fig. 4. Source data for all graphs is available as Excel file, Suppl. Data 1.

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

## Acknowledgements

We thank all members of the DMV lab, L. Mahadevan, P. Sens, E. Hannezo, for helpful discussions and Jorge Barbazan for help fibronectin depletion. We acknowledge the Cell and Tissue Imaging facility (PICT-IBiSA) and Animal Facility, Institut Curie. We thank Adeline Boyreau for her help with capsule imaging. The work is supported by the European Union's Horizon 2020 research and innovation program: European Research Council (ERC) under the grant agreement CoG 772487 (DMV) and StG 757557 (JLM), by Agence Nationale pour la Recherche (MecaTiss, ANR-17-CE30-0007-03, MecanoAdipo, ANR-17-CE13-0012-01) and Institut National du Cancer (PLBIO-2020-122) (PN), the European Molecular Biology Organization Young Investigator program (EMBO YIP) (JLM), Labex Cell(n)Scale (DMV) and Labex DEEP (ANR-11-LABX-0044) (JLM) which is part of the IDEX PSL (ANR-10-IDEX-0001-02). CL is supported by institutional grants from the Curie Institute, INSERM and CNRS, and by grants from Association Française contre les Myopathies (CAV-STRESS-MUS n°14293), Agence Nationale de la Recherche (MOTICAV ANR-17-CE13-0020-01), the Fondation ARC pour la Recherche sur le Cancer (Programme Labellisé PGA1-RF20170205456), and Institut National du Cancer (PLBIO-2018-08).

## Author contributions

F.B. conceptualized the study, initial observations, encapsulation, time-lapse imaging, immunostainings, capsule dissolving exp, micropipette and microplate experiments, pressure experiments, different cell types, analyzed data, assembled figures, wrote the first draft of the paper; C.U.M. non-adhesive substate exp, encapsulation, micropipette exp; L.A. encapsulation, depletion exp, immunostaining, different cell types; O.Z. depletion exp, non-adhesive substate exp, micropipette exp, western blots; different cell types; L.d.P. micropipette exp; M.D. and M.M.N. developed theory; A.R. encapsulation; K.A. and B.G.G. developed capsule technology; F.F. and A.A. microplate compression; C.L. supervised study; J.-L.M. supervised

micropipette exp, supervised the study, revised manuscript; P.N. conceptualized study, developed capsule technology, supervised the study, wrote the paper and revised manuscript; D.M.V. conceptualized the study, supervised the study, wrote the paper and revised manuscript.

## Competing interests

The authors declare no competing interests.
