## [Peer review file · Communications Biology]

Compressive stress triggers fibroblasts spreading over cancer cells to generate carcinoma in situ organizationReviewers' comments:

Reviewer #1 (Remarks to the Author):

This work by Bertillot, Martin, and colleagues describes what happens if cancer cells and stromal fibroblasts are co-cultured in a confined sphere. When space becomes limiting, the fibroblasts are observed to 'spread' around the outside of the cancer cells. This is correlated with fibronectin deposition, although the effect of fibronectin depletion is marginal and no causal link can be reliably inferred. A biophysical model is constructed to explain this phenomenon. Of note, this involves a role for the 'surface tension' of the fibroblasts. The experimental set up is unusual and interesting. The authors inter-disciplinary approach is innovative and commendable. However, the work is somewhat under-developed and it feels a bit premature for publication. Nonetheless, it could be well-suited to Comms Bio if substantial additional work were to be performed.

Specific comments

1. The work would be greatly strengthened if the authors had some way of altering the surface tension of the two cell types. It is likely that actomyosin may contribute to the surface tension and the authors should firstly test if this is the case (blebbistatin would be an easy way to start) and, if so, then perturb actomyosin in each cell type individually (MYL9 or MYH9 shRNA might be useful) and determine if this disrupts the phenomenon of compression induced spreading.
2. What is the elastic modulus of both cell types? Are the fibroblasts softer than the cancer cells? The reason for asking is that it seems quite conceivable that the softer cell type is effectively squashed when space becomes limiting and ends up in a thin layer between the inside of the sphere and the stiffer cell type. As I understand it, the model does not take into account the elastic modulus of the two cell types. It would be good if the authors could also explore this in their theoretical model.
3. The fibronectin data are inconclusive and don't warrant a position in the main figures. The shRNA data show a very small effect with no statistical significance and no control for off-target effects. No evidence is presented that it acts as a 'glue'. Have the authors considered adherens junctions as a possible 'glue'? What would cadherin or catenin depletion do?
4. The CT26 data are incomplete. First, the legend for SF1B is lacking. More critically, there is no data to indicate that they first adopt a side by side spheroid conformation and only transition to CT26 spreading when confinement becomes an issue. There is also nothing to suggest the fibronectin idea is relevant to CT26 cells. More generally, the inclusion of another isolate of CAFs would be beneficial. The authors could then compare cancer cell - CAF spheroids with different pairings, including contrasting epithelial and mesenchymal cancer cells.
5. The measurement of surface tension 30 minutes after dissolution of the capsule is sub-optimal. The cytoskeletal state of the cells can change a lot in this time frame. I can't really see a solution to this problem, but it should certainly be discussed.
6. How do the authors determine the correct value for interfacial tension to be used in their model?

Reviewer #2 (Remarks to the Author):

I think that this paper is very smart and demonstrates interesting model for study of the interactions between cancer cells and CAF in 3D-heterogeneous spheroids. The pictures are excellent and well support the interpretation of of results. I think that this apper will be of the great interst for people working with speroids/organoids and piblication will be useful. The materials and methods are well written that is important for repetition of experiemnt that is important for paper introducing the new meodel.I encourage to accept this manuscript in the present form.

Reviewer #3 (Remarks to the Author):

The authors employed an interesting assay that provides a spatial constraint in 3D to explore the spatial organisation of co-cultured cancer cells and cancer-associated fibroblasts. They found that

the two cell types segregate with fibroblasts enwrap the bulk of cancer cells in 3D sheet-like manner. They argue that their observed morphological features are due to compressive stress generated by cell proliferation. The study is interesting, but there are a couple of major concerns that need to be addressed prior to publication:

1) It is essential to see the generality of the main observations using other cancer and fibroblast cell types. Particularly the choice of human and mouse cell types seems quite random.

2) The relevance to in vivo conditions should be clear, and if necessary, the abstract/ introduction should be modified, or appropriate references should be cited to support the in vitro observations. Particularly authors claim that enwrapping of CAFs around the tumor is trivial; while they refer to the publications from their own group (refs 6 and 10), I cannot find any in vivo data supporting such enwrapment.

Other points:

-The method of making hollow alginate capsules should be appropriately shown/ described (a schematic of the technique in Figure 1 might be helpful).

-Fig 3C: either be fully moved to SI or appropriately described in the main text.

-Fig 4A: The details of measurements of pressure should be presented: I am curious to see the images showing the changes in the capsule's radius. Also, there is no clear explanation for why the pressure-time curve is linear at the start but non-linear after 20hr. Is this because of the non-linear effects on the alginate capsule or the type of non-linear pressure generated due to proliferation?

Reviewer #1 (Remarks to the Author):

This work by Bertillot, Martin, and colleagues describes what happens if cancer cells and stromal fibroblasts are co-cultured in a confined sphere. When space becomes limiting, the fibroblasts are observed to 'spread' around the outside of the cancer cells. This is correlated with fibronectin deposition, although the effect of fibronectin depletion is marginal and no causal link can be reliably inferred. A biophysical model is constructed to explain this phenomenon. Of note, this involves a role for the 'surface tension' of the fibroblasts. The experimental set up is unusual and interesting. The authors inter-disciplinary approach is innovative and commendable. However, the work is somewhat under-developed and it feels a bit premature for publication. Nonetheless, it could be well-suited to Comms Bio if substantial additional work were to be performed.

Specific comments

1. The work would be greatly strengthened if the authors had some way of altering the surface tension of the two cell types. It is likely that actomyosin may contribute to the surface tension and the authors should firstly test if this is the case (blebbistatin would be an easy way to start) and, if so, then perturb actomyosin in each cell type individually (MYL9 or MYH9 shRNA might be useful) and determine if this disrupts the phenomenon of compression induced spreading.

We thank the reviewer for their suggestions. We treated cancer cell and fibroblast spheroids with blebbistatin, an inhibitor of myosin II. As expected, this treatment lowered the surface tension of both, cancer cell's and fibroblast's, aggregates (**Fig. R1A**). However, upon treating co-cultured capsules with blebbistatin or Y-27632, a Rho-associated protein kinase (ROCK) inhibitor, we didn't observe fibroblast spreading before reaching confluence, nor did we note any hindrance to fibroblast spreading post-confluence (**Fig. R1B-D**).

Fig. R1. Inhibition of myosin

A. Individual (dots) surface tension measurements on cancer cells (red) and fibroblasts (green) spheroids treated with blebbistatin or DMSO (Control). One-way ANOVA test. n = 31 capsules. B. Percentage of capsules in which fibroblasts envelop spheroids of cancer cells over time. Capsules were either untreated Control (blue), or treated with blebbistatin (brown) and Y27632 (orange). Drugs were applied about 15h before the confluence. T=0 corresponds to the confluent stage. n=60 capsules. C. Confocal images of co-culture after fibroblast spreading, treated with blebbistatin or Y27632 (day 12). Fibroblasts expressed GFP (green), cancer cells are unstained, F-actin (phalloidin, magenta). Scale bars: 20 μ m.

Because these inhibitors affect both cancer cells and fibroblasts, and due to the inability to accurately determine their penetration efficiency through alginate capsules and, subsequently, the spheroids, we followed the reviewer's suggestion to specifically deplete Myh9 (myosin IIA) in either the cancer cells or the fibroblasts.

The targeted depletion of myosin IIA in cancer cells diminished the aggregate's surface tension (**Fig. 4G**), which noticeably delayed the spreading of fibroblasts. By day 4, only about 30% of capsules displayed fibroblast spreading (**Fig. 4H**), underlining the necessity of increased surface tension in cancer cell aggregates to trigger fibroblast spreading. We have included these findings on line 230.

Conversely, depleting myosin IIA from fibroblasts (**Suppl. Fig. 2B**) also delayed their spreading over wild-type cancer cells (**Suppl. Fig. 2D**), but the decrease in their surface tension was rather marginal (**Suppl. Fig. 2C**). This data implies that myosin IIA influences fibroblast motility, and its role in surface tension could potentially be compensated by other myosins, such as Myh10 (myosin IIB). We have included these findings on line 266.

Considering the possibility of surface tension modulation by other factors apart from actomyosin contractility, for instance, cell-cell adhesions or cell-matrix adhesions, we depleted fibronectin and N-cadherin in fibroblasts (**Fig. 3A, C**). Both these interventions lowered the surface tension of fibroblast aggregates (**Suppl. Fig. 2E, F**), but didn't induce fibroblast spreading before achieving the necessary pressure threshold at the confluence and instead delayed their spreading compared to the control cells (**Fig. 3B, D**). We have included these findings on lines, 163, 172, and 270.

Thus, we concluded that a straightforward application of the differential surface tension theory cannot explain the spatial arrangement of fibroblasts around cancer cells under confinement.

2. What is the elastic modulus of both cell types? Are the fibroblasts softer than the cancer cells? The reason for asking is that it seems quite conceivable that the softer cell type is effectively squashed when space becomes limiting and ends up in a thin layer between the inside of the sphere and the stiffer cell type. As I understand it, the model does not take into account the elastic modulus of the two cell types. It would be good if the authors could also explore this in their theoretical model.

The reviewer raises a very good point. Initially, we were intrigued by the observed enwrapping process and sought out a purely physical mechanism relying on the differential material properties of the cells. The apparent ability of fibroblasts to squeeze between the alginate wall and the tumor spheroid might suggest greater fibroblast deformability.

However, the term “deformable” could refer to an elastic property (marked by Young’s modulus) or a viscous one.

In fact, the fingering pattern we noticed was reminiscent of a Saffman-Taylor instability typically seen in a Hele-Shaw cell when a less viscous fluid is injected into a more viscous one (10.1098/rspa.1958.0085). We hypothesized that a difference in viscosity (rather than stiffness) between the two cell aggregates might elucidate the spreading behavior.

Nonetheless, we couldn't dismiss the intuitive hypothesis based on the differential stiffness proposed by the reviewer.

Additionally, we performed microplate experiments to assess the rheological properties of 3T3 and HT29 spheroids. This method estimates the surface tension by analyzing the spheroid's shape (radii of curvature), assuming it to be a liquid. Conversely, it can provide an estimate of Young’s modulus of the spheroid if one assumes that it behaves as a solid.

Young’s modulus E is derived from the measured force F at steady state, the contact area A between the plate and the spheroid, and the strain ϵ according to the equation: $E=F/(A.\epsilon)$. The results we obtained from approximately 30 spheroids of each type are depicted in **Fig. R2**. These data indicate that E_{3T3} (~660Pa) exceeds E_{HT29} (~440 Pa), which is opposite to the expected trend.

Fig. R2. Young’s modulus of cancer cells and fibroblasts aggregates

Moreover, we conducted frequency-dependent experiments by oscillating the microplate probe at a particular frequency (doi.org/10.1063/1.2202921). We restricted ourselves to one frequency, namely 1 Hz, on $n=3$ spheroids due to the time-intensive and challenging nature of these experiments. From the analysis of the phase shift and ratio between the excitation and the measured deformation signals, we derived the so-called loss modulus, G'' . At this frequency, $G''_{3T3} \sim 350 \text{ Pa} > G''_{HT29} \sim 200 \text{ Pa}$, meaning that, in disagreement with the abovementioned hypothesis that would support a Saffman-Taylor instability, NIH3T3 spheroids are more viscous than HT29 aggregates.

It should be noted that we assessed here the rheological properties of the multicellular aggregates, whereas the Reviewer refers to the stiffness of cells, which probably implicitly means individual cells. However, fibroblasts spread collectively, so the mechanical properties of the tissue as a whole (tumor aggregate or compressed fibroblast sheet) would

presumably be more relevant. As shown in other configurations, Young's modulus of epithelial sheets under stretch (doi: 10.1073/pnas.1213301109) or the compressibility of spheroids under osmotic shock (doi: 10.7554/eLife.63258) is strikingly different from the same parameters derived on individual cells, suggesting the potential dominance of ECM in the mechanical properties of cellular assemblies.

Regarding the reviewer's final point on incorporating cell stiffness into our theoretical model, we must clarify that the model is based on surface tension considerations and thus presumes the spheroids behave as liquids. Consequently, any parameters characterizing the spheroid's stiffness would not be applicable. The experiments cited above do not appear to support the initial intuition. It's worth noting that even though the liquid hypothesis might seem audacious, it forms the basis of the DAH (Differential Adhesion Hypothesis) and the timescale of spreading (several hours) corresponds to a typical ultra-low frequency (< 1 mHz), which is compatible with a liquid behavior. Ideally, a theoretical approach treating the spheroids as viscoelastic models would be helpful, but we are unaware of any such analytical framework.

We have now clarified this point added in the Supplementary Note, line 93.

3. The fibronectin data are inconclusive and don't warrant a position in the main figures. The shRNA data show a very small effect with no statistical significance and no control for off-target effects. No evidence is presented that it acts as a 'glue'. Have the authors considered adherens junctions as a possible 'glue'? What would cadherin or catenin depletion do?

We fully agree with the reviewer's criticism that the impact of fibronectin depletion was minor, statistically non-significant, and lacked off-target effect control. To address this point, we transfected cells individually with five distinct shRNAs, varying in efficacy, leading to varying levels of fibronectin depletion (**Fig. 3C**). When we encapsulated cancer cells with fibroblasts expressing different fibronectin levels, we observed that partial fibronectin depletion delayed spreading, while total depletion entirely inhibited fibroblast spreading (**Fig. 3D**). Thus, prior to reaching confluence, fibronectin might function as a "glue" between fibroblasts based on its localization and as documented in primary fibroblasts (doi: 10.1016/j.yexcr.2019.111616), but after confluence, it likely serves as an adhesive substrate facilitating fibroblast spreading.

Following the reviewer's suggestion, we also depleted N-cadherin (**Fig. 3A**), the main cadherin in fibroblasts, and observed that its absence similarly delayed fibroblast spreading (**Fig. 3B**). Collectively, we infer that both cell-cell and cell-matrix adhesions are required for the spreading of fibroblasts over cancer cells in confined conditions. We have included these findings starting on line 162.

4. The CT26 data are incomplete. First, the legend for SF1B is lacking. More critically, there is no data to indicate that they first adopt a side by side spheroid conformation and only transition to CT26 spreading when confinement becomes an issue. There is also nothing to suggest the fibronectin idea is relevant to CT26 cells. More generally, the inclusion of another isolate of CAFs would be beneficial. The authors could then compare cancer cell –

CAF spheroids with different pairings, including contrasting epithelial and mesenchymal cancer cells.

We apologize for the missing legend. We have now added a legend for **Fig. S1B** and incorporated both phase contrast and fluorescent images depicting the pre-confluent stage, which highlight that HT29 and CT26 cells adopt a side-by-side arrangement. It's evident from the post-confluence image that CT26 cells spread only after confinement.

We attempted to encapsulate additional CAFs from different patients but to no avail. Given that CAFs are primary non-immortalized cells, they exhibit particularly poor growth, especially as 3D aggregates. By the end of the experiment, which spans about 15 days, hardly any CAFs survive within the capsule. We speculated that the stark difference in proliferation rates between cancer cells and primary CAFs could be a contributing factor, leading us to adjust the cancer cell-to-CAF ratio at the encapsulation stage (increasing CAFs concentration 3 times). We also try to boost CAFs proliferation by adding growth factors, such as TGF- β and FGF, and coating the capsule with collagen. Unfortunately, even in these trials, CAFs had low survival rates. This finding hints at a competitive interaction between cancer cells and CAFs, which could be intriguing to explore in future studies, though we consider it beyond the scope of the current work.

Until now, we knew that mesenchymal cells (NIH3T3, CAFs, and CT26) adopt a side-by-side configuration with epithelial cells (HT29) prior to confluency and expand over their surface post-confluency. As suggested by the reviewer, we have now investigated whether this is a feature exclusive to mesenchymal and epithelial pairings. To this end, we encapsulated NIH3T3 cells (mesenchymal) with either CT26 cancer cells, which are fully mesenchymal, or HCT116 cancer cells, which exhibit an intermediate epithelial-mesenchymal phenotype. We observed that NIH3T3 and CT26 did not segregate post-encapsulation but remained intermixed, even following confinement. Conversely, NIH3T3 and HCT116 cells segregated and primarily maintained this segregation post-confinement, with a few NIH3T3 cells intermixed with the cancer cells. We did not see NIH3T3 cells spreading over the surface of these cancer cells. This data is now included in **Supp. Figure 1C, D** and described in the text, starting with line 110.

Collectively, these experiments indicate that the typical organization of carcinoma in situ can only be achieved with a combination of fully epithelial and fully mesenchymal cells. This also suggests that once cancer cells initiate the epithelial-to-mesenchymal transition, they begin to intermingle with fibroblasts, as observed in invasive carcinoma.

5. The measurement of surface tension 30 minutes after dissolution of the capsule is sub-optimal. The cytoskeletal state of the cells can change a lot in this time frame. I can't really see a solution to this problem, but it should certainly be discussed.

We fully agree with the Reviewer. Performing these surface tension measurements within 30 minutes after shell dissolution was already a technical achievement, but we cannot rule out significant cytoskeletal rearrangements that undoubtedly may affect surface tension. Even with this delay in assessing the surface tension of the released spheroids, we noticed differences compared to the non-compressed counterparts. Understandably, this difference

could potentially be more pronounced or even reversed if the experiments were executed more rapidly. Essentially, our data suggest a trend that supports enwrapping. Consequently, we have incorporated a sentence in the main text on line 214 to discuss this potential issue, and in the Methods section at line 524.

6. How do the authors determine the correct value for interfacial tension to be used in their model?

The model does not necessitate a precise value for the interfacial tension, γ_{fc} . This might be seen as a potential limitation. However, given the lack of a known experimental method to measure γ_{fc} , we opted to develop a model based on a phase diagram, where the energetically favorable configurations (side-by-side or core-shell) depend on the ratio of the two surface tensions (γ_f / γ_c) and the ratio of their difference to the interfacial tension ($(\gamma_f - \gamma_c) / \gamma_{fc}$). As illustrated in Suppl. Fig. 2A, when confinement is factored in, the region of the phase diagram where the side-by-side configuration is likely to occur significantly shrinks and determines the contact angle between the two spheroids. If we assume that γ_{fc} remains constant, the changes we observed in surface tension align with a movement (along a horizontal line) towards a region in the phase space where fibroblast enwrapping is the preferred configuration. We acknowledge that an experimental determination of γ_{fc} would allow us to accurately pinpoint the energetic state of the compressed cellular capsule in this phase diagram, providing a critical test for the model. Hence, our theoretical approach is currently used to understand the trend. We have clarified this in the text.

Reviewer #2 (Remarks to the Author):

I think that this paper is very smart and demonstrates an interesting model for the study of the interactions between cancer cells and CAF in 3D-heterogeneous spheroids. The pictures are excellent and well support the interpretation of results. I think that this paper will be of great interest for people working with spheroids/organoids and publication will be useful. The materials and methods are well written that is important for the repetition of the experiment that is important for the paper introducing the new model. I encourage to accept this manuscript in its present form.

We are grateful to the reviewer for their positive feedback on our research.

Reviewer #3 (Remarks to the Author):

The authors employed an interesting assay that provides a spatial constraint in 3D to explore the spatial organisation of co-cultured cancer cells and cancer-associated fibroblasts. They found that the two cell types segregate with fibroblasts enwrap the bulk of cancer cells in 3D sheet-like manner. They argue that their observed morphological features are due to compressive stress generated by cell proliferation. The study is interesting, but there are a couple of major concerns that need to be addressed prior to publication:

1) It is essential to see the generality of the main observations using other cancer and

fibroblast cell types. Particularly the choice of human and mouse cell types seems quite random.

We agree with the reviewer that it is important to show if this 3D model could be generated with other cell types. Please see a more detailed answer to the question from reviewer 1, point 4.

Briefly, our results show that mesenchymal cells (NIH3T3, CAFs, and CT26) adopt a side-by-side configuration with epithelial cells (HT29) prior to confluency and expand over their surface post-confluency. As suggested by reviewer 1, we have investigated whether this is a feature exclusive to mesenchymal and epithelial pairings. To this end, we encapsulated NIH3T3 cells (mesenchymal) with either CT26 cancer cells, which are fully mesenchymal, or HCT116 cancer cells, which exhibit an intermediate epithelial-mesenchymal phenotype. We observed that NIH3T3 and CT26 did not segregate post-encapsulation but remained intermixed, even following confinement. Conversely, NIH3T3 and HCT116 cells segregated and primarily maintained this segregation post-confinement, with a few NIH3T3 cells intermixed with the cancer cells. We did not see NIH3T3 cells spreading over the surface of these cancer cells. This data is now included in **Supp. figure 1** added their description in the text, line 110.

Altogether, our data imply that the typical structure of carcinoma in situ can only be recreated with a combination of fully epithelial and fully mesenchymal cells.

2) The relevance to in vivo conditions should be clear, and if necessary, the abstract/introduction should be modified, or appropriate references should be cited to support the in vitro observations. Particularly authors claim that enwrapping of CAFs around the tumor is trivial; while they refer to the publications from their own group (refs 6 and 10), I cannot find any in vivo data supporting such enwrapment.

Apologies for any previous lack of clarity. It's established in the literature that premalignant human tumors, or carcinoma in situ, across numerous human cancer types, are often encapsulated by fibrotic tissue, composed of fibroblasts and the extracellular matrix (ECM), (see examples: ref 6, Glentis et al, Suppl. Fig. 1; and newly added references such as Garcia-Vicien et al). However, these data points present a static view of tumor organization at a single time point. The dynamic process and mechanisms leading to this specific organization remain unknown. Observations of such fibrotic encapsulation have also been made in murine models, as shown in ref 10, Barbazan et al, Fig. 1a, b; and added references Staneva et al, Fig. 1a; Ozdemir et al. This suggests the feasibility of using mouse models to investigate the organization of fibroblast/ECM capsule in tumors. Nonetheless, the likely temporally slow nature of this process makes real-time studies of capsule formation technically challenging. Hence, the aim of our study was not to mimic the dynamic process of fibroblasts enwrapping cancer cells per se. Instead, we aimed to construct a 3D model that recapitulates the spatial organization of tumors observed in vivo. We have updated the abstract and introduction to include additional references, line 58 to fibrotic tumor encapsulation and to elucidate the primary objective of our study more explicitly.

Other points:

-The method of making hollow alginate capsules should be appropriately shown/ described (a schematic of the technique in Figure 1 might be helpful).

We have added a schematic representation of the experimental setup (**Fig. 1A**).

-Fig 3C: either be fully moved to SI or appropriately described in the main text.

We followed the suggestion of the Referee and move it to the **Supp. Figure 2A**.

-Fig 4A: The details of measurements of pressure should be presented: I am curious to see the images showing the changes in the capsule's radius. Also, there is no clear explanation for why the pressure-time curve is linear at the start but non-linear after 20hr. Is this because of the non-linear effects on the alginate capsule or the type of non-linear pressure generated due to proliferation?

This is indeed an interesting observation that we discussed in our previous work focused on the growth of encapsulated mono-cultures (doi: 10.1073/pnas.1309482110). Like the Reviewer, we first thought it could be due to a non-linear elasticity of the alginate capsule at large deformations. Thus, we had assessed the stress-strain relationship of the alginate capsules and found indeed a non-linear behavior for deformations >15%. We corrected this effect by adopting a classical phenomenological stress-strain dependence of the quadratic form: $\sigma = E(1 + A\varepsilon)\varepsilon$, where A is a constant to be fitted. In doing so, we have shown that the pressure could be reliably derived from the deformation measurements. The explanation we had proposed to account for this fast-growing pressure phase during the first 20h followed by the slower increase at later stages, is detailed in a simplified model at the end of the PNAS paper and then precisely investigated with agent-based simulations. In brief, the growth rate of the spheroid is not altered at all (i.e. there is no break in the growth curve) within the 20-24h following 3D confluence, suggesting that cells that have entered their cycle are not instantaneously affected by the compressive cue and will complete division. After a typical cell division time (1 day), all cells are compressed and start to reduce their proliferation rate, thus, the rate of volume increase inside the capsule.

We have now included a kymograph showing changes in the capsule's radius over time (new **Fig. 5B**).

Referee expertise:

Referee #1: cancer-associated fibroblasts, imaging, cancer progression, tumour microenvironment

Reviewers' comments:

Reviewer #1 (Remarks to the Author):

The revised manuscript from Vignjevic and colleagues is significantly improved from the previous version. Of note, the authors have now tested functional interference with N-Cad and MYH9, measured the mechanical properties of cells, and employed additional cancer cells. The rebuttal letter is very well-written and clarifies numerous issues. Overall, the work is now close to being suitable for publication.

Specific comments

1. The interesting new data shown in Figures R1 and R2 in the rebuttal should be incorporated into the manuscript, probably as supplementary figures. The lack of effect of blebbistatin is interesting because it argues that a non-surface tension mechanism is sufficient, but that imbalancing surface tension does disrupt the system. I note the authors' argument about drug penetration, but these are small molecules and other groups routinely use them in alginate systems. The authors should also be able to image the autofluorescence of blebbistatin.
2. The measurements of elasticity and viscosity are interesting and provide some further refinement to the picture of what is happening, including refuting my suggestion around elasticity.

Reviewer #1 (Remarks to the Author):

The revised manuscript from Vignjevic and colleagues is significantly improved from the previous version. Of note, the authors have now tested functional interference with N-Cad and MYH9, measured the mechanical properties of cells, and employed additional cancer cells. The rebuttal letter is very well-written and clarifies numerous issues. Overall, the work is now close to being suitable for publication.

We thank the reviewer for their valuable suggestions that we believe improved the overall quality of the manuscript.

Specific comments

1. The interesting new data shown in Figures R1 and R2 in the rebuttal should be incorporated into the manuscript, probably as supplementary figures. The lack of effect of blebbistatin is interesting because it argues that a non-surface tension mechanism is sufficient, but that imbalancing surface tension does disrupt the system. I note the authors' argument about drug penetration, but these are small molecules and other groups routinely use them in alginate systems. The authors should also be able to image the autofluorescence of blebbistatin.

In response to the reviewer's suggestion, we have incorporated Figure R1 into the Supplementary data as Figure 2 and have amended the text to elaborate on these observations as suggested, on line 226.

We acknowledge the reviewer's valid point that both blebbistatin and Y27, being small molecules, should penetrate alginate, which was previously shown (PMID: 36001958). Our observations confirm that, upon the addition of blebbistatin, fibroblasts change their shape, acquiring more wavy protrusions characteristic of myosin-inhibited cells.

However, we were uncertain regarding the extent of drugs' penetration into spheroids and their subsequent efficacy in inhibiting contractility in cancer cells.

The reviewer recommended imaging autofluorescence signals to assess the penetration depth of blebbistatin in our co-culture system. While we understand the rationale behind this suggestion, practical implementation raises substantial concerns. Blebbistatin's sensitivity to blue light (illumination below 500 nm), results in the molecule's photoconversion, inducing substantial cellular phototoxicity when imaged in living cells. The alternative, Para-Nitroblebbistatin, typically used for live experiments due to its lack of toxicity, is not autofluorescent.

Although we did not measure the penetration depth of blebbistatin, other studies indicate that the differential penetration depth of drugs into cancer spheroids is size-dependent (e.g., PMID: 31067739). According to this, it can be inferred that blebbistatin, being a small molecule, should penetrate the entire spheroid, corroborating our observation regarding its impact on the surface tension of cancer cell spheroids. However, given that blebbistatin reduces the surface tension of both cell types - cancer cells and fibroblasts - we agree with the reviewer that inhibiting contractility separately in each cell type is much more informative.

2. The measurements of elasticity and viscosity are interesting and provide some further refinement to the picture of what is happening, including refuting my suggestion around elasticity.

Even though the reviewer's intuition was refuted by our additional Young's modulus measurements, we agree that it deserves a specific comment and the corresponding data, now incorporated in Supp. Fig. 3A, might be of interest to readers. We have amended the text to elaborate on these observations on line 240.

REVIEWERS' COMMENTS:

Reviewer #2 (Remarks to the Author):

This is an interesting study and I commend the authors. The revisions have been implemented appropriately. I recommend prompt publication.